# ION-C: Integration of Overlapping Networks via Constraints

## Abstract

In many causal learning problems, variables of interest are often not all measured over the same observations, but are instead distributed across multiple datasets with overlapping variables. Tillman et al. (2008) presented the first algorithm for enumerating the minimal equivalence class of ground-truth DAGs consistent with all input graphs by exploiting local independence relations, called ION. In this paper, this problem is formulated as a more computationally efficient answer set programming (ASP) problem, which we call ION-C, and solved with the ASP system *clingo*. The ION-C algorithm was run on random synthetic graphs with varying sizes, densities, and degrees of overlap between subgraphs, with overlap having the largest impact on runtime, number of solution graphs, and agreement within the output set. To validate ION-C on real-world data, we ran the algorithm on overlapping graphs learned from data from two successive iterations of the European Social Survey (ESS), using a procedure for conducting joint independence tests to prevent inconsistencies in the input.

## 1 Introduction

Many inference problems require the use of data from different sources. Ideally, these data can be merged and collected into a single unified dataset (e.g., in tabular form) that is suitable for most learning methods. However, this type of data merging is not always possible. For example, suppose we have two distinct datasets, one from a financial institution and one from a healthcare provider. We might reasonably suspect that information about health outcomes and financial outcomes are related to one another; that is, we might want a unified model over these datasets. In practice, though, these datasets almost certainly cannot be integrated together for privacy reasons. Even worse, the datasets might be about different samples (even if from the same population), preventing us from directly linking observations from each dataset. At the same time, we might be able to leverage the variables that are measured in both datasets, such as someone's age, postal code, and so forth. We thus aim to learn about relationships between variables that are not co-measured in any dataset (existing or integrated), but where there are some variables that are measured in multiple datasets.

Formally, we examine a method for enumerating the complete set of ground-truth graphs $\mathcal{H}_i \in \mathbb{H}$ consistent with a set of input graphs $\mathcal{G}_i \in \mathbb{G}$, each learned locally from a source dataset.[1] The first algorithm for solving this problem, Integration of Overlapping Networks (ION) (Tillman et al., 2008), used a constructive solution that iterated through sets of changes to the complete graph that were faithful to independence relations in the input graphs. However, this formulation was computationally expensive, and only able to be tested on 4- and 6-node ground-truth graphs. In this work, we present a more efficient answer set programming formulation that, when solved, yields the same output set of graphs as ION; we call this ION-C, or ION via Constraints.

In Section 2, we describe previous approaches to learning from data distributed across datasets. Section 3 presents and explains the answer set programming formulation of ION-C. In Section 4, we provide evaluation results for ION-C for a range of synthetic input graphs. In Section 5, we evaluate ION-C on real-world data from two iterations of the European Social Survey. In Section 6, we discuss limitations and potential extensions of the ION-C algorithm.

---

[1]We assume that the "overlap graph" for $\mathcal{G}_i$ is connected; that is, for any $\mathcal{G}_j, \mathcal{G}_k$, there is a sequence of graphs from $\mathcal{G}_j$ to $\mathcal{G}_k$ such that each pair of graphs in the sequence have non-empty intersection of their variable sets.

## 2 RELATED WORK

Most structure learning methods (causal or otherwise) have focused on learning from a single dataset. As a result, there has been significant work on methods to unify datasets involving distinct variable sets (i.e., some variables are never co-measured) so that existing methods can be used. Most notably, since the 1960s, statistical matching approaches match individual observations from each dataset to observations from other datasets on the basis of distance in the *co-measured* features (Budd & Radner, 1969; Okner, 1972). These matches provide the basis for either imputations of unobserved variable values, or other statistical information connecting non-comeasured variables (Leulescu & Agafitei, 2013).

Traditional statistical matching approaches are only provably reliable when non-overlapping variables from each input dataset are conditionally independent of one another given the overlapping variables. More precisely, in the two dataset case where $\mathbf{D}_{1/2}$ is over $\mathbf{V}_{1/2} \cup \mathbf{V}_c$, these methods assume that $\mathbf{V}_1 \perp \mathbf{V}_2 | \mathbf{V}_c$. This assumption is both rarely true in practice, and also untestable given only the input datasets (Sims, 1972; Rodgers, 1984). While methods to overcome this conditional independence assumption exist, they usually require the provision of additional data (Paass, 1986; Singh et al., 1993), or the existence of informational proxy variables (Zhang, 2015).

Federated learning (FL) methods also aim to combine distinct information sources. In this case, we typically aim to learn a single model (at a central server) from multiple data sources, ideally without exchanging any observations and without assuming i.i.d. data across the different sources (Kairouz et al., 2021). In typical "horizontal" FL problems, each data source contains a partition of observations over a shared feature space; in "vertical" FL, data sources contain different features about shared observations (Wei et al., 2022). Some vertical FL methods also require sample alignment between data sources via cryptographic communication protocols (Lu & Ding, 2020). Federated *transfer* learning approaches aim to find single central models learned from information sources with both different sets of features and different observations, but typically with some small overlap in observations (Liu et al., 2020; Sharma et al., 2019).

While there are some high-level similarities, FL approaches inhabit a different problem space to the ION problem, since they seek efficient learning of a single best model rather than the full space of possible models given the data. They also are typically designed for distributed learning where a unified dataset could (in theory) be constructed. As a result, they usually face constraints of privacy and information flow that do not arise in our setting.

This paper is most directly related to Tillman et al. (2008), which presented an asymptotically correct algorithm that outputs the equivalence class of directed acyclic graphs (DAGs) consistent with an input set of partial ancestral graphs (PAGs). Their Integration of Overlapping Networks (ION) algorithm starts with a complete graph, then encodes edge absence and orientation information from each input PAG, including propagation of all entailments Zhang (2007). ION then finds all minimal sets of changes that would block paths between variables that are d-separated in at least one input PAG. These minimal changes are applied and propagated, and the resulting graph is accepted if it does not contradict the input PAGs. Finally, additional edge removals are tested to discover additional valid graphs. ION was shown to be both complete and sound, but is NP-complete and requires a superexponential number of operations. As a result, Tillman et al. (2008) were only able to run ION on 4- and 6-node ground-truth graphs.

ION takes PAGs as input; Tillman & Spirtes (2011) developed the Integration of Overlapping Datasets (IOD) algorithm that takes datasets as input. Their approach is closely related to the original ION algorithm, except that independence and association information is derived from p-value pooling over multiple datasets, rather than inferred from the input PAGs. IOD requires less memory than ION, and also outperformed ION in precision and recall, largely because IOD smoothly resolves (statistical) inconsistencies between input datasets.

Boolean satisfiability (SAT) solvers have also been applied to versions of this problem. Triantafillou et al. (2010) used a SAT solver to find a single graph that encodes all possible pairwise causal relationships between variables. Hyttinen et al. (2013) used a SAT formulation of d-separation to discover cyclic causal models from a set of overlapping input graphs.

Our approach uses answer set programming (ASP), a declarative problem-solving framework in which logical rules are provided to describe solution conditions for the problem (Marek &

Truszczyński, 1999; Gelfond & Lifschitz, 1988). Relative to other problem-solving methods, ASP benefits from a simple problem formulation and high expressiveness (Eiter et al., 2009; Brewka et al., 2011), while leveraging optimization of the boolean SAT problem (Gebser et al., 2007). ASP has been used to encode other causal learning problems Sonntag et al. (2015); Rantanen et al. (2020). For example, Hyttinen et al. (2014) used ASP to represent causal discovery as an optimization problem, providing a set of dependence and independence relations with weights corresponding to their probabilities, and returning the optimal causal graph according to these weights.

## 3 PROBLEM SETTING & METHOD

The problem that ION and ION-C aim to solve is to determine the complete set of ground-truth DAGs over all variables (that appear in at least one dataset) that are consistent with a set of overlapping input graphs. More formally: our inputs are a set of partial ancestral graphs (PAGs) $\mathcal{G}_i \in \mathbb{G}$, such that every graph $\mathcal{G}_i$ shares at least one node with at least one other graph in the set (and these overlaps for a connected structure; see footnote 1). Importantly, although all output graphs are DAGs, the input graphs do not have to be DAGs. In this problem, there are known latent variables for every input graph (namely, variables that are only in a different graph). Some of those latents could be common causes, which produce bidirected edges in the input PAG.

The output is a complete set of solution graphs $\mathbb{H}$, where each graph $\mathcal{H}_i \in \mathbb{H}$ is a DAG containing the union of all nodes in every input graph $\mathcal{G}_i$, such that each $\mathcal{H}_i$ does not violate any of the local independence or association information encoded in the input graphs. Specifically, this means that all d-separation and d-connection relations in every input graph $\mathcal{G}_i$ are preserved in every $\mathcal{H}_i$.

As a concrete example, suppose that $\mathcal{G}_1 = X \rightarrow Y \rightarrow Z$ and $\mathcal{G}_2 = X \rightarrow W \rightarrow Z$. Exactly two graphs (over $\{W, X, Y, Z\}$) preserve the d-separation and d-connection relations in these graphs: $\mathbb{H} = \{X \rightarrow Y \rightarrow W \rightarrow Z, X \rightarrow W \rightarrow Y \rightarrow Z\}$. Interestingly, in this example, we can learn that there must be a direct connection between $Y$ and $W$ (but not orientation of the edge), even though $Y$ and $W$ are never jointly measured.

In this paper, we present an answer set programming formulation of the integration of overlapping networks problem, which is implemented in the ASP system *clingo* (Gebser et al., 2019), based on the solver *clasp* (Gebser et al., 2007). We define the ION problem by providing the graph as a set of facts, then define a set of rules that must hold in any valid solution. *clingo* then outputs the set of all possible graphs that follows all of these facts and rules (see Listing 1).

The input PAGs are specified through sets of statements involving three different predicates:

1. `edge(X,Y,T).`, denoting an edge from node $X$ to node $Y$ in input PAG $T$

2. `bidirected(X,Y,T).`, denoting a bidirected edge between $X$ and $Y$ in $T$

3. `nedge(X,Y,T).`, denoting absence of an edge in either direction between $X$ and $Y$ in $T$

We additionally explicitly indicate all nodes in PAG $T$ with the command `varin(T, X).`. Finally, we provide the number of subgraphs and nodes as constants, and define all nodes with the command `node(0..n)`.

Listing 1 describes the problem specification in a format suitable for *clingo*. Line 1 defines any set of edge declarations between nodes as a valid solution. Lines 3 through 5 specify constraints for the solution: (3) self-loops are not allowed; (4) if an edge is absent in some input graph, then it cannot appear in a solution;[2] and (5) a valid solution must be acyclic. Lines 7 and 8 recursively define a directed path from $Y$ and $X$. Lines 10 and 11 provide a recursive definition of a directed edge from $X$ to $Y$ relative to the input graph $T$. Such an edge could be explained by a direct edge in the output graph, and also by a directed path that involves only nodes that do not appear in $T$ (since such a path would be an edge in $T$). Lines 13 and 14 define a causal connection between nodes $X$ and $Y$ in input graph $T$ as a directed edge between nodes, or an unobserved common cause of both nodes. Line 15 states that a bidirected edge in the input graph $T$ implies a causal connection between nodes without a directed edge in the solution graph, due to an unobserved common cause.

---

[2]Edge absence in an input graph indicates a d-separation (conditional independence) relation that must be preserved in all output graphs, and so the output DAGs also cannot have an edge.

Listing 1: *clingo* problem specification for ION-C problem.

```
1   {edge(X,Y)} :- node(X), node(Y).
2
3   :- edge(X,Y), X = Y.
4   :- edge(X,Y), nedge(X,Y,T), varin(T,X), varin(T,Y).
5   :- edge(X,Y), path(Y,X).
6
7   path(Y,X) :- edge(Y,X).
8   path(Y,X) :- edge(Y,Z), path(Z,X).
9
10  directed(X,Y,T) :- edge(X,Y), varin(T,Y).
11  directed(X,Y,T) :- edge(X,Z), directed(Z,Y,T), not varin(T,Z).
12
13  causalconn(X,Y,T) :- directed(X,Y,T).
14  causalconn(X,Y,T) :- directed(Z,X,T), directed(Z,Y,T), not varin(T,Z).
15  bidirected(X,Y,T) :- causalconn(X,Y,T), not directed(X,Y,T).
16
17  :- nedge(X,Y,T), causalconn(X,Y,T), varin(T,X), varin(T,Y).
18  :- edge(X,Y,T), not directed(X,Y,T), varin(T,X), varin(T,Y).
19
20  #show edge/2.
```

Line 17 specifies that the nonexistence of an edge (either directed or bidirected) between two nodes in the same input graph $T$ implies the lack of a causal connection. Line 18 specifies the converse: a directed edge between two nodes in the same input graph implies a directed path between them. Finally, line 19 specifies the output of edge pairs for all solution graphs.

In order to show that the ION-C ASP formulation leads to the correct output equivalence class, we show that the problem statement is complete and sound.

**Theorem 3.1.** *Soundness: If nodes $X$ and $Y$ are d-separated (d-connected) given nodes $\mathbf{Z}$ in some $\mathcal{G}_i \in \mathbb{G}$, then $X$ and $Y$ are d-separated (d-connected) given $\mathbf{Z}$ in every output $\mathcal{H}_i \in \mathbb{H}$.*

*Proof.* Suppose $X$ and $Y$ are d-separated given $\mathbf{Z}$ in some $\mathcal{G}_i$, but d-connected in some output $\mathcal{H}_i$. This implies that there is a path between $X$ and $Y$ in $\mathcal{H}_i$ that is active given $\mathbf{Z}$. $X$ and $Y$ are not adjacent in $\mathcal{G}_i$, and so (by line 17) the output graph d-connection cannot be a directed path or common cause. The only remaining possibility is that some variable in $R \in \mathbf{Z}$ is a descendant of a collider in $\mathcal{H}_i$ on a path between $X$ and $Y$. This implies, however, that $\mathcal{H}_i$ includes paths from $X$ to $R$ and $Y$ to $R$ that are active given $\mathbf{Z}$ $R$. However, this implies (per lines 10-11) that each of these paths corresponds to a sequence of edges in $\mathcal{G}_i$ that contradict the known d-separation in $\mathcal{G}_i$.

Now suppose that $X$ and $Y$ are d-connected given $\mathbf{Z}$. Line 18 specifies that if an edge exists between two nodes $X$ and $Y$ in input graph $T$, then the property `directed(X,Y,T)` is true. Per lines 10 and 11, `directed(X,Y,T)` holds true only when there is an edge from $X$ to $Y$ in the output, or when the solution includes multiple edges from $X$ to $Y$ consisting of intermediate nodes that were not observed in graph $T$. This means that any pair of nodes connected by an edge in an input $T$ will be connected either by a single edge, or by a directed path of nodes that were not included in $T$. This, in turn, entails the necessary d-connection relation. $\square$

**Theorem 3.2.** *Completeness: Let $\mathcal{H}_i$ be a partial ancestral graph over variables $\mathcal{V}$ such that for every $\{(X,Y)\} \subseteq \mathcal{V}$, if $X$ and $Y$ are d-separated (d-connected) given $\mathbf{Z} \subseteq \mathcal{V}/\{X,Y\}$ in some $\mathcal{G}_i \in \mathbb{G}$, then $X$ and $Y$ are d-separated (d-connected) given $\mathbf{Z}$ in $\mathcal{H}_i$. Then, $\mathcal{H}_i$ is in $\mathbb{H}$.*

*Proof.* In order to show completeness, we must show that no d-separations or d-connections present in the input graph are unnecessarily removed from the output set $\mathbb{H}$. All edge removals in line 4 are necessary to translate d-separations from the inputs, as is the acyclicity constraint in line 5. Remaining edge removals only occur in line 14 and 15 by removing bidirected edges $X \leftrightarrow Y$ and retaining the relevant d-connections by creating directed paths to $X$ and $Y$ from the unobserved common cause, or in line 11 to replace a directed edge $X \rightarrow Y$ with a previously unobserved path

of edges $X \rightarrow Z \rightarrow Y$. Because *clingo* outputs the entire set of solution graphs matching the given constraints, and because none of the changes specified by these constraints would preclude such an output $\mathcal{H}_i$ from the solution set, ION-C is complete for the problem. $\quad\square$

## 4 SIMULATION RESULTS

In Tillman et al. (2008), the ION algorithm was only evaluated on 4- and 6-node directed acyclic graphs (DAGs) due to computational constraints. In order to establish the usability of the ION-C algorithm on larger graphs with the faster ASP formulation (and additional computational resources), we tested ION-C on graphs of varying sizes, densities, and overlap between subgraphs.

We randomly generated "ground truth" graphs using four control parameters: (i) the total number of nodes $\mathcal{N}$; (ii) $p_{degree}$ that controls ground-truth density; (iii) the number of input subgraphs $s$; and (iv) $p_{overlap}$ that controls the extent of input graph overlap. More precisely, each ground-truth graph was generated with $\mathcal{N}$ nodes, and random edges such that each node makes connections to $a$ other nodes, with $a \sim \text{Bin}(\mathcal{N} - 1, p_{degree})$. As $p_{degree}$ increases, more connections are made, and ground-truth graphs are denser. Finally, we check that the DAG is connected, and add required edges to connect the graph if not. To generate input subgraphs, we first split the nodes evenly into $s$ partitions, and for each partition set, we sample $p_{overlap}$ of the nodes from other partitions. As $p_{overlap}$ increases, each subgraph will contain more nodes, and the level of overlap between subgraphs will increase.

Given the ground-truth graph and a subset of nodes, we analytically generate the input subgraph by marginalizing out the variables not in the subset. The resulting input PAG is provably causally faithful to the ground-truth. For example, if the ground-truth contains $X \rightarrow Z \rightarrow Y$ but the subgraph does not include $Z$, then the input PAG will have $X \rightarrow Y$. In addition, we connect nodes $X \leftrightarrow Y$ if they share a common cause that is not observed in that subgraph. Given a set of input PAGs for a single ground-truth graph, we convert the inputs (as described in Section 3) and run the ASP solver to find the full set of possible ground-truth graphs consistent with the input graphs.

We ran 100 simulated ground-truth graphs for each possible combination of parameters, with $\mathcal{N} \in \{6, 8, 10, 15, 25\}$, $p_{overlap} \in \{0.25, 0.5, 0.75\}$, $p_{degree} \in \{0.1, 0.25, 0.5, 0.75\}$, and $S \in \{2, 3, 4\}$. For graphs with 15 and 25 nodes, due to the high complexity of denser graphs, we additionally used $p_{degree}$ values of 0.025, 0.05, and 0.075. In total, we considered 234 sets of 100-graph simulations. All instances were run with four-hour timeouts for the *clingo* solver on nodes with 24 GB of RAM. We only report results for parameterizations that resulted in at least 95 of 100 ground-truths completing (and all reported proportions are relative to the completed runs). 153 parameterizations resulted in completion of at least 95 of 100 output solution sets.

For each simulation, we initially report two key statistics. First, *prop_same* is the proportion of all possible edges or edge absences that are shared across 75%, 90%, and 100% of the solution set. This statistic provides a measure of the similarity of graphs in the solution set. Second, *prop_accurate* indicates, as a proportion of the edges/absences shared in 75%, 90%, or 100% of the solution set, what proportion are found in the ground-truth graph itself (ignoring orientation). This statistic provides a measure of the "accuracy" of the output set: are the most common edges/absences correct? Complete results for all parameterizations are available in Appendix A. Figures 1, 2, and 3 show these statistics for all 8-node graphs, for which all graphs ran at all parameterizations. Tables 1 and 2 display *prop_same* and *prop_accurate* for completed parameterizations among 15- and 25-node graphs with two subgraphs.

As expected, the most important factor controlling the number of output graphs, and consequently the runtime of the algorithm, was the amount of overlap between the input subgraphs. For example, in 8-node ground-truth graphs with $p_{degree} = 0.75$ with two subgraphs (the rightmost set of bars in the left graph in Figure 3), the three settings of overlap corresponded to two subgraphs with 5, 6, and 7 nodes each. The median number of solution graphs was 25648, 161, and 5, respectively. (In many settings with $p_{overlap} = 0.75$, there was only one valid solution graph.) The degree of overlap in the graphs is also the largest factor in the coherence of the output set; as Figure 1 indicates, proportion of edge adjacencies or absences that is shared across 90% of the solution set is closely related to the overlap in nodes.

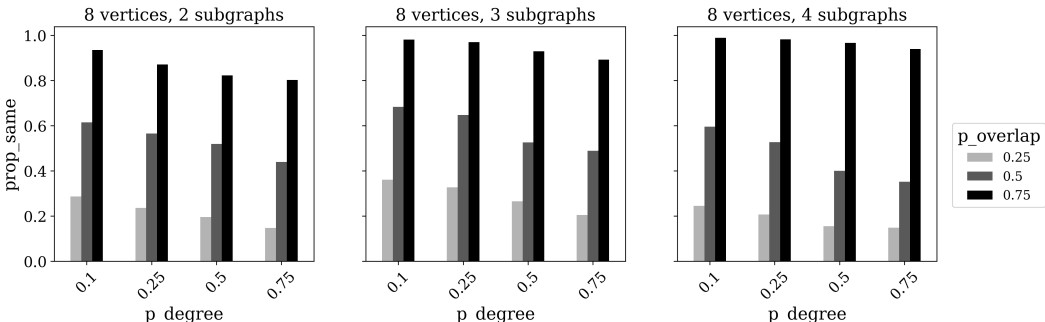

Figure 1: Mean proportion of edge adjacencies and absences shared in 90% of the solution set.

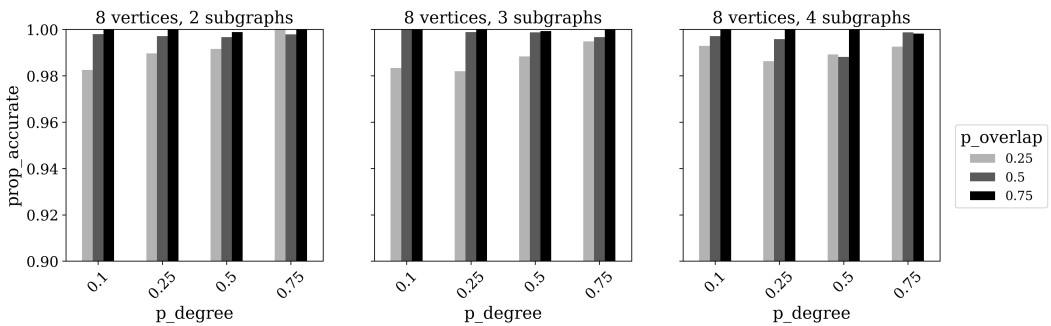

Figure 2: Mean proportion (of edge adjacencies and absences shared in 90% of the solution set) that match ground truth.

Lower overlap settings typically led to lower accuracy in terms of the widely-shared edges in the output set, though this was not the case in every parameterization run (see Figure 2). The number of input subgraphs that the ground truth was split into, $s$, had little impact compared to the degree of overlap, with similar results for 2, 3, and 4 subgraphs in these results. These patterns are replicated across all numbers of nodes tested, although with larger graphs, the simulations with $p_{degree} \geq 0.25$ are rarely reported because too many simulations timed out.

Input graphs with increased ground-truth density had on average larger solution sets across all numbers of vertices and subgraphs. We also observe slight decreases in the proportion of edges and edge absences shared in 90% of solutions as density increased, although in testing with larger graphs on lower densities, this decrease did not occur until density reached at least $p_{degree} = 0.1$.

Figure 3 reports the *median* number of graphs in the solution set; note that we have median of 1 output graph for many settings of $p_{overlap}$. Nonetheless, almost all parameterizations produced a very long tail in terms of runtime. Among all successful parameterizations we examined, the median ratio of the maximum runtime of successful graphs to the median runtime across all graphs was 10.58; the median ratio of the maximum runtime to the 90th-percentile runtime was 3.53. For example, in simulations with 15 nodes split into two subgraphs, $p_{degree} = 0.05$, and $p_{degree} = 0.25$: half of the graphs yielded solutions within 1.41 seconds; 90% finished within 161 seconds; but one graph (generated from the same parameters) took over 3.6 hours to solve.

In these simulations, we use the proportion of accurate edges and edge absences among those shared in a certain proportion of the solution set as a measure of confidence in each edge commission or omission (in Figures 1 and 2 that proportion is 90%.) On average, across every complete solution we examined, the average proportion of accurate edges or edge absences among those in at least 75% of solution graphs was 97.33%. When the threshold is increased to 90%, the average proportion increases to 99.55%. Edges that appear in 100% of solution set graphs were always accurate, as the input graphs are derived analytically (and ION-C is provably sound). However, as solution sets get larger, the proportion of shared edges or edge absences consistently decreases.

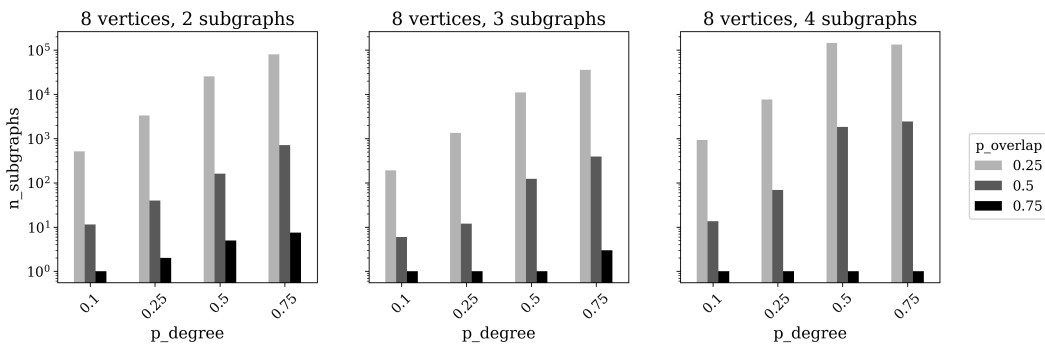

Figure 3: Median number of graphs in the solution set.

Table 1: For 15-node ground-truth graphs split in 2 subgraphs: Proportion of edges & absences found in $\geq 90\%$ of outputs (left); proportion of these edges & absences found in ground truth (right)

| | | $p_{overlap}$ | | | | | $p_{overlap}$ | | |
| --- | --- | --- | --- | --- | --- | --- | --- | --- | --- |
| | | **0.25** | **0.5** | **0.75** | | | **0.25** | **0.5** | **0.75** |
| | **0.025** | 0.382 | 0.696 | 0.945 | | **0.025** | 0.969 | 0.994 | 1.000 |
| | **0.050** | 0.385 | 0.706 | 0.947 | | **0.050** | 0.968 | 0.991 | 1.000 |
| $p_{degree}$ | **0.075** | * | 0.704 | 0.955 | $p_{degree}$ | **0.075** | * | 0.999 | 1.000 |
| | **0.100** | * | 0.726 | 0.953 | | **0.100** | * | 0.996 | 1.000 |
| | **0.250** | * | * | 0.921 | | **0.250** | * | * | 1.000 |
| | **0.500** | * | * | 0.829 | | **0.500** | * | * | 0.999 |
| | **0.750** | * | * | 0.805 | | **0.750** | * | * | 0.997 |

Table 2: For 25-node ground-truth graphs split in 2 subgraphs: Proportion of edges & absences found in $\geq 90\%$ of outputs (left); proportion of these edges & absences found in ground truth (right)

| | | $p_{overlap}$ | | | | $p_{overlap}$ | |
| --- | --- | --- | --- | --- | --- | --- | --- |
| | | **0.50** | **0.75** | | | **0.50** | **0.75** |
| | **0.025** | 0.705 | 0.921 | | **0.025** | 0.995 | 0.991 |
| $p_{degree}$ | **0.050** | 0.685 | 0.915 | $p_{degree}$ | **0.050** | 0.996 | 1.000 |
| | **0.075** | * | 0.928 | | **0.075** | * | 0.999 |
| | **0.100** | * | 0.917 | | **0.100** | * | 1.000 |

# 5 APPLICATION TO REAL-WORLD DATA

In order to examine the real-world performance and utility of ION-C, we use data from rounds 8 and 9 of the European Social Survey (ESS), from years 2016 and 2018, respectively (ERIC, 2017; 2019). The ESS survey, conducted every two years, asks participants a core set of questions in every survey, in addition to a rotating set of topical modules that vary in each iteration. Rotating modules not asked in the same survey round are thus not co-measured, but ION-C can potentially be used to enumerate possible ground-truth graphs based on graphs learned within each survey round.

We selected 8 variables from the "welfare attitudes" module from ESS round 8; 8 from the "justice and fairness" module from ESS round 9; and an overlap group of 8 variables that were measured in both survey rounds. We suspected that there might be connections between participants' attitudes about the round-specific topics; for example, someone who is particularly concerned about fairness might plausibly want a strong, supportive welfare system.

We learn causal graphs for each survey round using the PC algorithm (Spirtes et al., 2001), allowing for missing data using the method in Tu et al. (2019) implemented in the *causal-learn* Python package (Zheng et al., 2024). (Missing values correspond to nonresponses, refusals, and other non-answer codes from the ESS dataset.) In order to maintain consistency in causal structures among

the overlapping nodes, we use the p-value pooling method for testing independence across multiple datasets outlined in Algorithm 1 of Tillman & Spirtes (2011), and adjust the graphs in the same fashion as the synthetic graphs – this time, with no knowledge of the actual ground truth, but using the merged graph provided by the shared independence tests – and pass the two resulting graphs into the ION program.

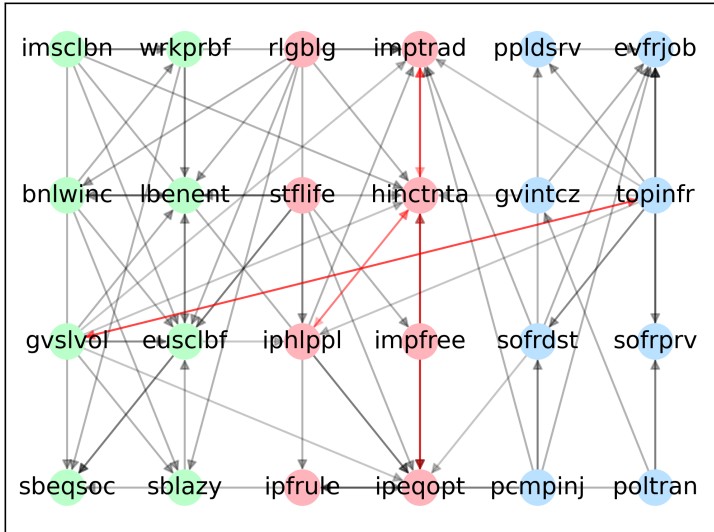

Figure 4: Representation of ION solution set.

The resulting ION-C solution set contained 2,046 graphs. Figure 4 displays the ION-C solution set, with edge opacity corresponding to the proportion of solution graphs that contain that edge. (Green (blue) nodes are variables only in ESS 8 (ESS 9); red nodes are those measured in both surveys. Full variable names are provided in Appendix B.) Edges that were not present in either input graph appear in red. Note that edges that appear bidirected in Figure 4 are not actually bidirected, but rather represent connections where different solution graphs orient the edge in different directions. Note also that not all edges that appear in an input graph appear in the entire output.

In total, 58 of 66 edges contained in the original graphs were present in all solution graphs, while the remainder all appeared in exactly 1,550 graphs. Meanwhile, edges not present in any input graph were present on average in 34.4% of solution graphs, although this does not merge edges in opposite directions between the same nodes.

We observe two kinds of added edges: those between nodes in the intersection of the inputs, and one pair of nodes that were not co-measured. This latter pair of nodes was *gvslvol*, a question in which participants were asked whether the standard of living of the elderly was the government's responsibility, and the other was *topinfr*, a question asking participants how fair the salaries of the top 10% of income earners was. This edge was observed in 1,984 of 2,046 solution graphs, with 992 graphs each containing this edge in each direction, making it the most common solution set edge not contained in either input graph. Moreover, this edge is arguably intuitively plausible, as both factors are related to people's high-level views about the role of government in economic support.

## 6 DISCUSSION

While the output of the ION-C algorithm is provably correct—that is, it returns all possible ground-truth graphs consistent with the input—there are limitations to this approach as a methods of causal discovery given overlapping graphs. Just as with the constructive formulation of ION in Tillman et al. (2008), contradictory information in input datasets, whether due to differences in underlying distributions in the data or statistical errors in the causal discovery process, can make the constraint formulation unsatisfiable, with no possible ground truths satisfying this conflict. The number of

conditional independence tests required in the PC algorithm is potentially super-exponential in the number of variables, and therefore the likelihood of mistaken edge commissions, deletions, or orientations drastically increases as dimensionality increases.

Not only can statistical errors lead to unsatisfiable ION-C problems, but if statistical errors occur in multiple input graphs, it is possible for ION-C to return a solution set that, while valid for the input graphs as stated, is inaccurate to the ground truth. Potential methods for improving such errors include the p-value pooling approach outlined in Tillman & Spirtes (2011), which ensures consistency in the causal structures over the overlapping nodes. Another option is to find the closest satisfiable set of graphs to the input set, using a metric like the structural Hamming distance (Tsamardinos et al., 2006) to compare to the original input. This latter approach will find valid ground-truths that require the fewest changes to the provided input graphs, even if the learned causal graphs are inconsistent with each other.

An additional limitation is in the interpretation of the output equivalence class of graphs. As seen in the results, these sets can range into the tens or hundreds of millions of graphs, even given relatively small input graphs. Of course, these large output sets are still much smaller than the super-exponential number of $n$-node DAGs, but large output sets might have limited real-world utility.

In this paper, we use the proportion of the solutions in which a given edge or edge absence appears as a sort of ad-hoc confidence metric; for example, a node that appears in 90% of the solution set is very likely to be present in the ground truth. This is not entirely baseless – ION provides all possible graphs consistent with the input, and barring input errors, the actual ground-truth is one of these graphs. Therefore, if we start with a flat prior over possible global graphs, then this measure accurately describes the likelihood of output graphs in our beliefs.

Indeed, in our results, we found that edges or edge absences that were in large proportions of the output set were very likely to be accurate. However, in order to more clearly determine the single ground truth, additional information or experiments would be needed to disambiguate ION-C solution set graphs. In this way, ION-C could serve to indicate edges of interest that are likely, but not certain to exist, or indicate edges that the solution set has high disagreement over, allowing an intervention on these edges to most efficiently cut down the set of possible ground truths as part of an experimental process. In Section 5, for example, we saw that the ION-C output, with a solution size in the thousands, involves disagreement over only a small number of edges, highlighting which variables and relationships we do not currently have the information to understand.

Even without leveraging other information, there are potentially other methods or assumptions that could help to deal with the size the ION-C solution set. To provide one example, suppose two potential ground-truth graphs $\mathcal{H}_1$ and $\mathcal{H}_2$ are returned by ION-C, where the edges in $\mathcal{H}_1$ are a proper subset of those in $\mathcal{H}_2$. We might make a simplicity assumption that leads us to focus on $\mathcal{H}_1$, the graph with fewer causal connections. In this fashion, by leveraging additional assumptions or requirements from the data, we can take the often very large solution set returned by ION-C and reduce it into more useful constructs for analysis.

### REPRODUCIBILITY STATEMENT

In order to reproduce the results described above, we provide the *clingo* code for the ION-C problem in Listing 1, and as part of supplementary material. In addition, all code used to conduct the simulations from Section 4, as well as code to output the ION problem given data from the ESS, is provided as part of supplementary material. Full results from the simulations we ran are available in Appendix A.

### AUTHOR CONTRIBUTIONS

Removed for anonymization

### ACKNOWLEDGMENTS

Removed for anonymization

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

# A FULL RESULTS ON SYNTHETIC DATA

Key for column headers:

- $\mathcal{N}$: number of vertices in ground truth
- $p_{degree}$: controls density of ground truth
- $p_{overlap}$: controls overlap of subgraphs
- $s$: number of subgraphs ground truth is split into
- Graphs Run: number of simulations in which ION-C returned the full solution set without timing out
- Runtime: median amount of time in which ION-C returned the full solution set
- Solution Graphs: median number of graphs in the ION-C solution set
- $S_{75}$: proportion of edges or edge absences shared in 75% of solution graphs ($prop\_same$)
- $A_{75}$: Among edges or edge absences shared in 75% of solution graphs, proportion that are accurate to ground truth ($prop\_accurate$)
- $S_{90}$: proportion of edges or edge absences shared in 90% of solution graphs
- $A_{90}$: Among edges or edge absences shared in 75% of solution graphs, proportion that are accurate to ground truth
- $S_{100}$: proportion of edges or edge absences shared in 100% of solution graphs, all of which are accurate

Table 3: Full results of simulations on synthetic graphs.

| $\mathcal{N}$ | $p_{degree}$ | $p_{overlap}$ | $s$ | Graphs Run | Runtime | Solution Graphs | $S_{75}$ | $A_{75}$ | $S_{90}$ | $A_{90}$ | $S_{100}$ |
|---|---|---|---|---|---|---|---|---|---|---|---|
| 6 | 0.1 | 0.25 | 2 | 100 | 0.018 | 24 | 0.534 | 0.912 | 0.362 | 1 | 0.332 |
| 6 | 0.1 | 0.25 | 3 | 100 | 0.027 | 480 | 0.331 | 0.888 | 0.119 | 0.975 | 0.055 |
| 6 | 0.1 | 0.25 | 4 | 100 | 0.022 | 130 | 0.393 | 0.864 | 0.208 | 0.968 | 0.149 |
| 6 | 0.1 | 0.5 | 2 | 100 | 0.017 | 1 | 0.887 | 0.997 | 0.871 | 1 | 0.871 |
| 6 | 0.1 | 0.5 | 3 | 100 | 0.017 | 7.5 | 0.709 | 0.947 | 0.575 | 0.992 | 0.561 |
| 6 | 0.1 | 0.5 | 4 | 100 | 0.018 | 3 | 0.767 | 0.98 | 0.692 | 0.995 | 0.684 |
| 6 | 0.1 | 0.75 | 2 | 100 | 0.018 | 1 | 1 | 1 | 1 | 1 | 1 |
| 6 | 0.1 | 0.75 | 3 | 100 | 0.014 | 1 | 0.964 | 0.993 | 0.938 | 1 | 0.938 |
| 6 | 0.1 | 0.75 | 4 | 100 | 0.015 | 1 | 0.98 | 0.997 | 0.97 | 1 | 0.97 |
| 6 | 0.25 | 0.25 | 2 | 100 | 0.016 | 27.5 | 0.561 | 0.929 | 0.383 | 0.997 | 0.333 |
| 6 | 0.25 | 0.25 | 3 | 100 | 0.028 | 494.5 | 0.318 | 0.865 | 0.142 | 0.944 | 0.076 |
| 6 | 0.25 | 0.25 | 4 | 100 | 0.031 | 275.5 | 0.371 | 0.879 | 0.181 | 0.966 | 0.119 |
| 6 | 0.25 | 0.5 | 2 | 100 | 0.013 | 2 | 0.859 | 0.991 | 0.825 | 1 | 0.821 |
| 6 | 0.25 | 0.5 | 3 | 100 | 0.019 | 9 | 0.655 | 0.966 | 0.531 | 0.997 | 0.512 |
| 6 | 0.25 | 0.5 | 4 | 100 | 0.015 | 6 | 0.712 | 0.986 | 0.618 | 1 | 0.602 |
| 6 | 0.25 | 0.75 | 2 | 100 | 0.016 | 1 | 1 | 1 | 1 | 1 | 1 |
| 6 | 0.25 | 0.75 | 3 | 100 | 0.016 | 1 | 0.939 | 0.994 | 0.915 | 1 | 0.913 |
| 6 | 0.25 | 0.75 | 4 | 100 | 0.014 | 1 | 0.961 | 0.997 | 0.951 | 1 | 0.951 |
| 6 | 0.5 | 0.25 | 2 | 100 | 0.02 | 70.5 | 0.592 | 0.943 | 0.317 | 0.998 | 0.278 |
| 6 | 0.5 | 0.25 | 3 | 100 | 0.031 | 1341.5 | 0.32 | 0.894 | 0.099 | 0.995 | 0.059 |
| 6 | 0.5 | 0.25 | 4 | 100 | 0.025 | 645 | 0.381 | 0.898 | 0.137 | 0.957 | 0.088 |
| 6 | 0.5 | 0.5 | 2 | 100 | 0.017 | 3 | 0.846 | 0.988 | 0.761 | 0.999 | 0.753 |
| 6 | 0.5 | 0.5 | 3 | 100 | 0.019 | 22 | 0.644 | 0.944 | 0.446 | 0.989 | 0.413 |
| 6 | 0.5 | 0.5 | 4 | 100 | 0.017 | 14.5 | 0.685 | 0.952 | 0.524 | 1 | 0.501 |
| 6 | 0.5 | 0.75 | 2 | 100 | 0.015 | 1 | 1 | 1 | 1 | 1 | 1 |
| 6 | 0.5 | 0.75 | 3 | 100 | 0.015 | 1 | 0.921 | 0.992 | 0.85 | 1 | 0.846 |
| 6 | 0.5 | 0.75 | 4 | 100 | 0.015 | 1 | 0.946 | 0.996 | 0.934 | 1 | 0.934 |
| 6 | 0.75 | 0.25 | 2 | 100 | 0.02 | 112.5 | 0.565 | 0.954 | 0.308 | 0.995 | 0.265 |
| 6 | 0.75 | 0.25 | 3 | 100 | 0.034 | 1529 | 0.32 | 0.922 | 0.079 | 0.995 | 0.035 |
| 6 | 0.75 | 0.25 | 4 | 100 | 0.026 | 622.5 | 0.407 | 0.956 | 0.153 | 0.992 | 0.109 |

| | | | | | | | | | | | |
|---|---|---|---|---|---|---|---|---|---|---|---|
| 6 | 0.75 | 0.5 | 2 | 100 | 0.018 | 3 | 0.868 | 0.988 | 0.767 | 1 | 0.76 |
| 6 | 0.75 | 0.5 | 3 | 100 | 0.02 | 30 | 0.653 | 0.98 | 0.426 | 0.998 | 0.402 |
| 6 | 0.75 | 0.5 | 4 | 100 | 0.017 | 14.5 | 0.692 | 0.982 | 0.536 | 0.998 | 0.516 |
| 6 | 0.75 | 0.75 | 2 | 100 | 0.017 | 1 | 1 | 1 | 1 | 1 | 1 |
| 6 | 0.75 | 0.75 | 3 | 100 | 0.017 | 1 | 0.91 | 0.991 | 0.847 | 1 | 0.842 |
| 6 | 0.75 | 0.75 | 4 | 100 | 0.019 | 1 | 0.938 | 0.996 | 0.923 | 1 | 0.922 |
| 8 | 0.1 | 0.25 | 2 | 100 | 0.041 | 513 | 0.474 | 0.905 | 0.287 | 0.982 | 0.176 |
| 8 | 0.1 | 0.25 | 3 | 100 | 0.038 | 190 | 0.518 | 0.933 | 0.361 | 0.983 | 0.265 |
| 8 | 0.1 | 0.25 | 4 | 100 | 0.054 | 932 | 0.422 | 0.904 | 0.245 | 0.993 | 0.161 |
| 8 | 0.1 | 0.5 | 2 | 100 | 0.024 | 11.5 | 0.716 | 0.962 | 0.615 | 0.998 | 0.588 |
| 8 | 0.1 | 0.5 | 3 | 100 | 0.025 | 6 | 0.768 | 0.967 | 0.683 | 1 | 0.666 |
| 8 | 0.1 | 0.5 | 4 | 100 | 0.028 | 13.5 | 0.68 | 0.965 | 0.596 | 0.997 | 0.564 |
| 8 | 0.1 | 0.75 | 2 | 100 | 0.021 | 1 | 0.944 | 1 | 0.935 | 1 | 0.934 |
| 8 | 0.1 | 0.75 | 3 | 100 | 0.02 | 1 | 0.981 | 1 | 0.981 | 1 | 0.981 |
| 8 | 0.1 | 0.75 | 4 | 100 | 0.017 | 1 | 0.988 | 1 | 0.988 | 1 | 0.988 |
| 8 | 0.25 | 0.25 | 2 | 100 | 0.071 | 3346 | 0.449 | 0.927 | 0.236 | 0.99 | 0.134 |
| 8 | 0.25 | 0.25 | 3 | 100 | 0.053 | 1343 | 0.502 | 0.923 | 0.327 | 0.982 | 0.245 |
| 8 | 0.25 | 0.25 | 4 | 100 | 0.123 | 7685 | 0.391 | 0.895 | 0.207 | 0.986 | 0.127 |
| 8 | 0.25 | 0.5 | 2 | 100 | 0.039 | 40 | 0.687 | 0.961 | 0.566 | 0.997 | 0.526 |
| 8 | 0.25 | 0.5 | 3 | 100 | 0.04 | 12 | 0.744 | 0.978 | 0.648 | 0.999 | 0.612 |
| 8 | 0.25 | 0.5 | 4 | 100 | 0.047 | 69 | 0.648 | 0.97 | 0.527 | 0.996 | 0.497 |
| 8 | 0.25 | 0.75 | 2 | 100 | 0.016 | 2 | 0.906 | 0.991 | 0.871 | 1 | 0.867 |
| 8 | 0.25 | 0.75 | 3 | 100 | 0.042 | 1 | 0.971 | 1 | 0.969 | 1 | 0.969 |
| 8 | 0.25 | 0.75 | 4 | 100 | 0.023 | 1 | 0.987 | 0.999 | 0.982 | 1 | 0.982 |
| 8 | 0.5 | 0.25 | 2 | 99 | 0.257 | 25648 | 0.439 | 0.933 | 0.195 | 0.992 | 0.115 |
| 8 | 0.5 | 0.25 | 3 | 100 | 0.123 | 11016 | 0.465 | 0.942 | 0.265 | 0.988 | 0.198 |
| 8 | 0.5 | 0.25 | 4 | 99 | 1.511 | 144744 | 0.354 | 0.917 | 0.155 | 0.989 | 0.103 |
| 8 | 0.5 | 0.5 | 2 | 100 | 0.052 | 161 | 0.671 | 0.967 | 0.519 | 0.997 | 0.462 |
| 8 | 0.5 | 0.5 | 3 | 100 | 0.039 | 122.5 | 0.685 | 0.965 | 0.526 | 0.999 | 0.493 |
| 8 | 0.5 | 0.5 | 4 | 100 | 0.058 | 1847.5 | 0.552 | 0.958 | 0.4 | 0.988 | 0.341 |
| 8 | 0.5 | 0.75 | 2 | 100 | 0.018 | 5 | 0.872 | 0.991 | 0.823 | 0.999 | 0.81 |
| 8 | 0.5 | 0.75 | 3 | 100 | 0.017 | 1 | 0.939 | 0.998 | 0.929 | 0.999 | 0.926 |
| 8 | 0.5 | 0.75 | 4 | 100 | 0.019 | 1 | 0.969 | 1 | 0.966 | 1 | 0.964 |
| 8 | 0.75 | 0.25 | 2 | 100 | 0.704 | 80074 | 0.422 | 0.952 | 0.147 | 1 | 0.093 |
| 8 | 0.75 | 0.25 | 3 | 99 | 0.361 | 35907 | 0.461 | 0.948 | 0.204 | 0.995 | 0.146 |
| 8 | 0.75 | 0.25 | 4 | 100 | 1.268 | 133298 | 0.359 | 0.945 | 0.148 | 0.993 | 0.101 |
| 8 | 0.75 | 0.5 | 2 | 100 | 0.031 | 709 | 0.631 | 0.974 | 0.439 | 0.998 | 0.383 |
| 8 | 0.75 | 0.5 | 3 | 100 | 0.03 | 392.5 | 0.644 | 0.977 | 0.489 | 0.997 | 0.446 |
| 8 | 0.75 | 0.5 | 4 | 100 | 0.07 | 2455 | 0.548 | 0.973 | 0.352 | 0.999 | 0.313 |
| 8 | 0.75 | 0.75 | 2 | 100 | 0.021 | 7.5 | 0.866 | 0.99 | 0.802 | 1 | 0.794 |
| 8 | 0.75 | 0.75 | 3 | 100 | 0.021 | 3 | 0.917 | 0.996 | 0.892 | 1 | 0.886 |
| 8 | 0.75 | 0.75 | 4 | 100 | 0.021 | 1 | 0.953 | 0.997 | 0.939 | 0.998 | 0.936 |
| 10 | 0.1 | 0.25 | 2 | 100 | 0.023 | 144 | 0.637 | 0.946 | 0.511 | 0.987 | 0.449 |
| 10 | 0.1 | 0.25 | 3 | 100 | 0.231 | 18180 | 0.433 | 0.905 | 0.249 | 0.972 | 0.126 |
| 10 | 0.1 | 0.25 | 4 | 99 | 1.595 | 122920 | 0.405 | 0.91 | 0.215 | 0.976 | 0.094 |
| 10 | 0.1 | 0.5 | 2 | 100 | 0.017 | 6.5 | 0.816 | 0.986 | 0.77 | 1 | 0.76 |
| 10 | 0.1 | 0.5 | 3 | 100 | 0.022 | 9 | 0.773 | 0.979 | 0.715 | 0.999 | 0.702 |
| 10 | 0.1 | 0.5 | 4 | 100 | 0.02 | 13 | 0.758 | 0.98 | 0.698 | 0.998 | 0.672 |
| 10 | 0.1 | 0.75 | 2 | 100 | 0.017 | 1 | 0.957 | 0.997 | 0.943 | 1 | 0.943 |
| 10 | 0.1 | 0.75 | 3 | 100 | 0.018 | 1 | 0.996 | 1 | 0.995 | 1 | 0.995 |
| 10 | 0.1 | 0.75 | 4 | 100 | 0.018 | 1 | 0.985 | 1 | 0.983 | 1 | 0.983 |
| 10 | 0.25 | 0.25 | 2 | 100 | 0.053 | 2094 | 0.584 | 0.959 | 0.444 | 0.992 | 0.372 |
| 10 | 0.25 | 0.5 | 2 | 100 | 0.02 | 37 | 0.781 | 0.979 | 0.694 | 0.998 | 0.663 |
| 10 | 0.25 | 0.5 | 3 | 100 | 0.028 | 283 | 0.679 | 0.973 | 0.594 | 0.996 | 0.554 |
| 10 | 0.25 | 0.5 | 4 | 100 | 0.024 | 102 | 0.705 | 0.975 | 0.617 | 0.997 | 0.578 |
| 10 | 0.25 | 0.75 | 2 | 100 | 0.018 | 3 | 0.919 | 0.996 | 0.907 | 1 | 0.906 |
| 10 | 0.25 | 0.75 | 3 | 100 | 0.017 | 1 | 0.99 | 1 | 0.989 | 1 | 0.989 |
| 10 | 0.25 | 0.75 | 4 | 100 | 0.018 | 1 | 0.97 | 1 | 0.969 | 1 | 0.969 |
| 10 | 0.5 | 0.25 | 2 | 98 | 3.825 | 232272.5 | 0.524 | 0.947 | 0.347 | 0.989 | 0.27 |

| | | | | | | | | | | | |
|---|---|---|---|---|---|---|---|---|---|---|---|
| 10 | 0.5 | 0.5 | 2 | 100 | 0.047 | 965 | 0.679 | 0.977 | 0.566 | 0.996 | 0.521 |
| 10 | 0.5 | 0.5 | 3 | 98 | 0.433 | 34347.5 | 0.579 | 0.956 | 0.437 | 0.994 | 0.385 |
| 10 | 0.5 | 0.5 | 4 | 99 | 0.781 | 56920 | 0.565 | 0.955 | 0.434 | 0.995 | 0.37 |
| 10 | 0.5 | 0.75 | 2 | 100 | 0.02 | 8 | 0.902 | 0.996 | 0.88 | 1 | 0.876 |
| 10 | 0.5 | 0.75 | 3 | 100 | 0.02 | 2 | 0.941 | 0.998 | 0.929 | 1 | 0.928 |
| 10 | 0.5 | 0.75 | 4 | 100 | 0.02 | 2 | 0.948 | 0.998 | 0.935 | 1 | 0.934 |
| 10 | 0.75 | 0.25 | 2 | 100 | 42.265 | 3248227.5 | 0.496 | 0.947 | 0.294 | 0.989 | 0.224 |
| 10 | 0.75 | 0.5 | 2 | 100 | 0.18 | 11239 | 0.654 | 0.974 | 0.542 | 0.993 | 0.479 |
| 10 | 0.75 | 0.5 | 3 | 95 | 1.406 | 105252 | 0.581 | 0.963 | 0.433 | 0.995 | 0.375 |
| 10 | 0.75 | 0.5 | 4 | 100 | 3.142 | 207902 | 0.546 | 0.959 | 0.406 | 0.994 | 0.357 |
| 10 | 0.75 | 0.75 | 2 | 100 | 0.024 | 14.5 | 0.885 | 0.993 | 0.844 | 0.999 | 0.828 |
| 10 | 0.75 | 0.75 | 3 | 100 | 0.024 | 4 | 0.921 | 0.999 | 0.904 | 1 | 0.901 |
| 10 | 0.75 | 0.75 | 4 | 100 | 0.023 | 6.5 | 0.907 | 0.997 | 0.883 | 0.999 | 0.871 |
| 15 | 0.025 | 0.25 | 2 | 99 | 1.211 | 69616 | 0.607 | 0.91 | 0.382 | 0.969 | 0.214 |
| 15 | 0.025 | 0.5 | 2 | 100 | 0.031 | 29 | 0.767 | 0.975 | 0.696 | 0.994 | 0.661 |
| 15 | 0.025 | 0.5 | 3 | 100 | 0.039 | 96 | 0.726 | 0.971 | 0.644 | 0.996 | 0.606 |
| 15 | 0.025 | 0.5 | 4 | 100 | 0.037 | 20 | 0.794 | 0.985 | 0.733 | 0.997 | 0.702 |
| 15 | 0.025 | 0.75 | 2 | 100 | 0.022 | 1 | 0.952 | 0.998 | 0.945 | 1 | 0.94 |
| 15 | 0.025 | 0.75 | 3 | 100 | 0.022 | 1 | 0.988 | 0.997 | 0.978 | 1 | 0.978 |
| 15 | 0.025 | 0.75 | 4 | 100 | 0.023 | 1 | 0.989 | 1 | 0.988 | 1 | 0.988 |
| 15 | 0.05 | 0.25 | 2 | 99 | 1.413 | 122586 | 0.574 | 0.901 | 0.385 | 0.968 | 0.221 |
| 15 | 0.05 | 0.5 | 2 | 100 | 0.026 | 44 | 0.76 | 0.968 | 0.706 | 0.991 | 0.677 |
| 15 | 0.05 | 0.5 | 3 | 100 | 0.037 | 204 | 0.708 | 0.968 | 0.618 | 0.993 | 0.575 |
| 15 | 0.05 | 0.5 | 4 | 100 | 0.039 | 38.5 | 0.782 | 0.974 | 0.704 | 0.997 | 0.678 |
| 15 | 0.05 | 0.75 | 2 | 100 | 0.02 | 1 | 0.952 | 0.999 | 0.947 | 1 | 0.944 |
| 15 | 0.05 | 0.75 | 3 | 100 | 0.022 | 1 | 0.983 | 0.998 | 0.977 | 1 | 0.977 |
| 15 | 0.05 | 0.75 | 4 | 100 | 0.025 | 1 | 0.982 | 0.998 | 0.98 | 1 | 0.98 |
| 15 | 0.075 | 0.5 | 2 | 99 | 0.033 | 70 | 0.763 | 0.981 | 0.704 | 0.999 | 0.67 |
| 15 | 0.075 | 0.5 | 3 | 100 | 0.058 | 316 | 0.697 | 0.971 | 0.623 | 0.993 | 0.557 |
| 15 | 0.075 | 0.5 | 4 | 100 | 0.036 | 36 | 0.778 | 0.972 | 0.726 | 0.986 | 0.69 |
| 15 | 0.075 | 0.75 | 2 | 100 | 0.022 | 1 | 0.963 | 0.999 | 0.955 | 1 | 0.954 |
| 15 | 0.075 | 0.75 | 3 | 100 | 0.025 | 1 | 0.971 | 0.997 | 0.964 | 1 | 0.964 |
| 15 | 0.075 | 0.75 | 4 | 100 | 0.026 | 1 | 0.99 | 1 | 0.988 | 1 | 0.988 |
| 15 | 0.1 | 0.5 | 2 | 100 | 0.04 | 71.5 | 0.778 | 0.981 | 0.726 | 0.996 | 0.686 |
| 15 | 0.1 | 0.5 | 3 | 99 | 0.057 | 1624 | 0.671 | 0.968 | 0.601 | 0.996 | 0.546 |
| 15 | 0.1 | 0.5 | 4 | 98 | 0.059 | 162.5 | 0.749 | 0.982 | 0.69 | 0.996 | 0.651 |
| 15 | 0.1 | 0.75 | 2 | 100 | 0.026 | 1 | 0.958 | 0.998 | 0.953 | 1 | 0.95 |
| 15 | 0.1 | 0.75 | 3 | 100 | 0.028 | 1 | 0.974 | 1 | 0.971 | 1 | 0.971 |
| 15 | 0.1 | 0.75 | 4 | 100 | 0.03 | 1 | 0.985 | 1 | 0.984 | 1 | 0.984 |
| 15 | 0.25 | 0.75 | 2 | 100 | 0.025 | 5.5 | 0.929 | 0.999 | 0.921 | 1 | 0.919 |
| 15 | 0.25 | 0.75 | 3 | 100 | 0.03 | 4 | 0.928 | 0.998 | 0.919 | 1 | 0.917 |
| 15 | 0.25 | 0.75 | 4 | 100 | 0.03 | 2 | 0.944 | 0.999 | 0.941 | 1 | 0.94 |
| 15 | 0.5 | 0.75 | 2 | 99 | 0.065 | 1090 | 0.855 | 0.993 | 0.829 | 0.999 | 0.819 |
| 15 | 0.5 | 0.75 | 3 | 95 | 0.151 | 5328 | 0.827 | 0.994 | 0.799 | 0.999 | 0.789 |
| 15 | 0.5 | 0.75 | 4 | 97 | 0.061 | 688 | 0.852 | 0.994 | 0.833 | 0.999 | 0.821 |
| 15 | 0.75 | 0.75 | 2 | 98 | 0.348 | 11293.5 | 0.833 | 0.994 | 0.805 | 0.997 | 0.78 |
| 25 | 0.025 | 0.5 | 2 | 99 | 0.124 | 579 | 0.769 | 0.974 | 0.705 | 0.995 | 0.646 |
| 25 | 0.025 | 0.5 | 3 | 100 | 0.14 | 1166 | 0.75 | 0.978 | 0.682 | 0.993 | 0.621 |
| 25 | 0.025 | 0.5 | 4 | 100 | 0.137 | 448 | 0.782 | 0.979 | 0.72 | 0.995 | 0.68 |
| 25 | 0.025 | 0.75 | 2 | 100 | 0.062 | 3 | 0.937 | 0.988 | 0.921 | 0.991 | 0.909 |
| 25 | 0.025 | 0.75 | 3 | 100 | 0.064 | 2 | 0.952 | 0.999 | 0.944 | 1 | 0.94 |
| 25 | 0.025 | 0.75 | 4 | 100 | 0.072 | 1 | 0.987 | 0.999 | 0.985 | 1 | 0.984 |
| 25 | 0.05 | 0.5 | 2 | 95 | 0.431 | 7616 | 0.743 | 0.98 | 0.685 | 0.996 | 0.642 |
| 25 | 0.05 | 0.5 | 3 | 97 | 0.48 | 17080 | 0.721 | 0.982 | 0.666 | 0.995 | 0.615 |
| 25 | 0.05 | 0.5 | 4 | 95 | 0.17 | 1252 | 0.766 | 0.984 | 0.719 | 0.994 | 0.682 |
| 25 | 0.05 | 0.75 | 2 | 99 | 0.051 | 6 | 0.925 | 0.998 | 0.915 | 1 | 0.911 |
| 25 | 0.05 | 0.75 | 3 | 98 | 0.066 | 1 | 0.961 | 0.996 | 0.957 | 0.998 | 0.951 |
| 25 | 0.05 | 0.75 | 4 | 97 | 0.069 | 1 | 0.987 | 1 | 0.986 | 1 | 0.984 |
| 25 | 0.075 | 0.75 | 2 | 100 | 0.059 | 5 | 0.933 | 0.999 | 0.928 | 0.999 | 0.922 |

| 25 | 0.075 | 0.75 | 3 | 100 | 0.07 | 2 | 0.959 | 1 | 0.955 | 1 | 0.953 |
| 25 | 0.075 | 0.75 | 4 | 100 | 0.069 | 1 | 0.982 | 0.999 | 0.979 | 1 | 0.978 |
| 25 | 0.1 | 0.75 | 2 | 100 | 0.063 | 10 | 0.921 | 0.998 | 0.917 | 1 | 0.912 |
| 25 | 0.1 | 0.75 | 3 | 100 | 0.067 | 4 | 0.943 | 0.997 | 0.937 | 1 | 0.932 |
| 25 | 0.1 | 0.75 | 4 | 100 | 0.073 | 2 | 0.974 | 1 | 0.974 | 1 | 0.973 |

# B    EUROPEAN SOCIAL SURVEY VARIABLES

Table 4: Description of variables included in analysis of European Social Survey in Section 5 (ERIC, 2017; 2019)

| | survey | description |
|---|---|---|
| **rlgblg** | both | Belonging to particular religion or denomination |
| **stflife** | both | How satisfied with life as a whole |
| **iphlppl** | both | Important to help people and care for others well-being |
| **ipfrule** | both | Important to do what is told and follow rules |
| **imptrad** | both | Important to follow traditions and customs |
| **hinctnta** | both | Household's total net income, all sources (decile) |
| **impfree** | both | Important to make own decisions and be free |
| **ipeqopt** | both | Important that people are treated equally and have equal opportunities |
| **imsclbn** | ESS8 only | When should immigrants obtain rights to social benefits/services |
| **bnlwinc** | ESS8 only | Social benefits only for people with lowest incomes |
| **gvslvol** | ESS8 only | Standard of living for the old, governments' responsibility |
| **sbeqsoc** | ESS8 only | Social benefits/services lead to a more equal society |
| **wrkprbf** | ESS8 only | Benefits for parents to combine work and family even if means higher taxes |
| **lbenent** | ESS8 only | Many with very low incomes get less benefit than legally entitled to |
| **eusclbf** | ESS8 only | Against or In favour of European Union-wide social benefit scheme |
| **sblazy** | ESS8 only | Social benefits/services make people lazy |
| **ppldsrv** | ESS9 only | By and large, people get what they deserve |
| **gvintcz** | ESS9 only | Government in country takes into account the interests of all citizens |
| **sofrdst** | ESS9 only | Society fair when income and wealth is equally distributed |
| **pcmpinj** | ESS9 only | Convinced that in the long run people compensated for injustices |
| **evfrjob** | ESS9 only | Everyone in country fair chance get job they seek |
| **topinfr** | ESS9 only | Top 10% full-time employees in country, earning more than [amount], how fair |
| **sofrprv** | ESS9 only | Society fair when people from families with high social status enjoy privileges |
| **poltran** | ESS9 only | Decisions in country politics are transparent |

