# OpenReview forum: "ION-C: Integration of Overlapping Networks via Constraints"
_ICLR.cc/2025/Conference — Submitted to ICLR 2025_

### Official Review · Reviewer_HWkX · 2024-10-31

**Soundness:** 3
**Presentation:** 2
**Contribution:** 2
**Rating:** 3
**Confidence:** 4

**Summary:**

The authors formulate the problem of learning a causal graph from observation distributed across multiple datasets with overlapping variables as an Answer Set Programming problem for computational efficiency. Given a set of partial ancestral graphs obtained from multiple datasets separately, the authors specify the problem in the ASP system clingo and utilize an ASP solver called clasp to output the set of DAGs that is consistent with all the constraints given by the PAGs. The contribution is mainly based on the problem formulation and proving the soundness and completeness of the approach. Overall, the motivation of the work is strong, and the proposed method is intuitive. However, the overall content of the paper can be improved by carefully proofreading it, including more discussions on why the proposed method is preferred over the existing work, some background information on causal graphical models, assumptions used etc., and including a more thorough comparison in the experiment such as including the memory usage, adding synthetic experiments by directly using the synthetic samples to generate the inputs via causal discovery algorithms instead of using ground truth PAGs.

**Strengths:**

- Formulating the problem of integrating networks with overlapping variables as an answer set programming problem is interesting.
- The proposed approach outputs the correct equivalence class and the proofs seem correct.
- The paper provides extensive experiments, including both synthetic and real-world data.

**Weaknesses:**

- It’s not clear to me how the formulation alone is an improvement upon the previous work. I would suggest making it more explicit why this formulation is desirable over other existing work.
- The motivating example does not seem correct.
- The writing needs improvment.  For example, $D_{1/2}, V_{1/2}, V_{c}$ are not defined when it is mentioned the first time in the related work. It also impedes my understanding of the proofs of theorems. For details, please see my questions.
- I find these lines 155-157 very difficult to understand until I read the second part of the proof of theorem 3.1: "Such an edge could be explained by a direct edge in the output graph, and also by a directed path that involves only nodes that do not appear in T (since such a path would be an edge in T).”
- In the real-world experiment, the exact total number of variables used in lines 370-371 is difficult to determine. I need to read Figure 4 to understand it.

**Questions:**

- Line 066: What are $D_{1/2}, V_{1/2}, V_{c}$?
- Why should one prefer ASP formulation over the existing work, especially IOD?
- After reading the paper about IOD [1], the IOD paper uses ION for more than 6 nodes in comparison, the manuscript only describes IOD as something very similar to ION, but it seems like both ION and IOD can handle a larger set of nodes? Why are they not compared in the experiment?
- Based on line 249, the proposed method takes 24 GB to run all the experiments, is this suggesting ASP solver is memory intensive? As reported by the IOD paper, it has shown a case where it only uses 100 MB on average for 13 node cases (see Figure 1b on that paper). Shouldn’t memory usage also be compared for a fair comparison?
- Since the input is a set of PAGs, why should one output a set of DAGs in the end after integrating? Why is it correct to convert a bidirected edge to a directed edge?
- Lines 154-155: why lines 10-11 is described specifically "relative to the input graph T” while lines 7 and 8 are not?
- By looking at the proof of theorem 3.1, lines 10 and 11 seem to be ensuring d-separation statements hold in the output, is it true?
- Line 132: Why the complete set of solution graphs $\mathbb{H}$ does not contain some other possible DAGs e.g. $X\leftarrow Y \rightarrow W \rightarrow Z$ ? I think you should have 6 elements in the set since each edge can go both directions and one can subtract the possibilities of having $\rightarrow Y \leftarrow$ or $\rightarrow W \leftarrow$.
- Line 195: "so (by line 17) the output graph d-connection cannot be a directed path or common cause”, what do the authors mean here in the proof of Theorem 3.1?
- Line 198: “are active given $\mathbf{Z} R$", do the authors mean given $\mathbf{Z}$?
- Lines 201-202: "Per lines 10 and 11, directed(X,Y,T) holds true only when there is an edge from X to Y in the output
- Lines 233-234: "Finally, we check that the DAG is connected, and add required edges to connect the graph if not.”, Do you mean “if the DAG is connected”?
- Line 246: here $S$ is capital, is it supposed to be lowercase $s$ as in line 229?
- Would the authors mind explaining the advantages of integrating networks over using causal discovery methods in the presence of missing data? I imagine the latter is preferred when the overlapping variable set is small and the graph is slightly larger than 15 nodes.

---

> ### Author Response · Authors · 2024-11-20
>
> Thank you for the detailed feedback. We have clarified many of the points you indicated in your questions in the text. To go over some specific issues:
>
> > The motivating example does not seem correct.
>
> The motivating example has been made more clear to demonstrate hypothetical constraints ION-C might be useful for. In the case of finance and healthcare datasets, we might want to ask causal questions such as the relationship between financial stress and certain health outcomes. We may not necessarily be able to combine both datasets at the individual level for privacy reasons, as neither the financial or healthcare institution can share specific observations. However, if we are able to independently learn causal graphs from both datasets, assuming that we can safely share dataset-level causal information, then the ION approach allows us to interrogate those causal questions without operating at the data level.
>
> > Why should one prefer ASP formulation over the existing work, especially IOD?
>
> ASP solvers benefit from optimization made in conflict-driven SAT solving, while allowing the problem to be specified incredibly simply, as we show in Listing 1. Existing work in causal learning has indicated that these lead to significant runtime improvements for ASP approaches (Hyttinen et al., 2014; Sonntag et al., 2015). In addition, ASP approaches allow existing knowledge/beliefs about the ground-truth to be easily specified with simple constraints, whether soft or hard.
>
> > Since the input is a set of PAGs, why should one output a set of DAGs in the end after integrating? Why is it correct to convert a bidirected edge to a directed edge?
>
> We assume that the underlying ground truth is a DAG, and that in each input PAG to ION-C, bidirected edges occur only due to unobserved common causes in that input’s subset of variables from the DAG. When all variables are integrated into a single graph, these bidirected edges are no longer necessary, as the common cause is now present in the complete graph.

---

> > ### Comment · Reviewer_HWkX · 2024-11-21
> > **Thank you.**
> >
> > I thank the reviewer for the response. Some questions are not answered. After reading other reviews, I decided to keep my score.

---

### Official Review · Reviewer_k6i8 · 2024-11-02

**Soundness:** 3
**Presentation:** 3
**Contribution:** 2
**Rating:** 6
**Confidence:** 3

**Summary:**

The paper presents ION-C, a strategy to integrate different causal graphs from datasets with overlapping variables. In doing so, the authors extend the existing ION algorithm (Tillman et al., 2018) which, despite being sound and complete, has a faster-than-exponential complexity. ION-C instead tackles the problem by using logic programming, in particular Answer Set Programming, which they also prove to be sound and complete. They then test their approach on both simulated and real-world data.

**Strengths:**

The tackled problem is of practical importance as exploiting data from different sources, even when defined on different variables, is fundamental for real-world applications of causal discovery. Due to the high-computational complexity, the existing ION algorithm could not scale to medium-sized graphs — as the authors say ION was only tested on 6 nodes DAGs in the original evaluation. Therefore, the presented solution might have important applications for larger graphs.

**Weaknesses:**

The main point of the proposed ION-C algorithm is to fasten the ION algorithm. However, the computational complexity of solving the ASP program is not reported in the paper. If the authors could provide such complexity and compare it directly to the complexity of the original ION algorithm, it would help to understand whether ION-C has theoretical guarantees of being faster or if it's only an empirical result. Finally, for the experimental side, it would help to have a clear visualization of ION vs ION-C in terms of execution time for growing number of nodes in the graph.

**Questions:**

- What is the computational cost of solving an ASP problem?
- Are all the results in the experimental section from ION-C without a direct comparison with ION?

---

> ### Author Response · Authors · 2024-11-20
>
> Thank you for the feedback on better demonstrating the runtime and complexity improvements of ION vs. ION-C. The overall problem is NP-complete, and the structure of the problem makes it difficult to determine expected-case complexity. Evidence from previous problems indicates that ASP formulations can lead to significant speedups in causal learning, due to both optimization in conflict-driven solving and limited need for manual optimization of an algorithm.  We have introduced visualizations of runtime and expanded the discussion of it further (compared to the original manuscript).

---

> > ### Comment · Reviewer_k6i8 · 2024-11-26
> >
> > I thank the authors for their comment, and I will maintain my evaluation.

---

### Official Review · Reviewer_7BsB · 2024-11-03

**Soundness:** 4
**Presentation:** 3
**Contribution:** 2
**Rating:** 3
**Confidence:** 4

**Summary:**

This paper considers the problem of eliciting causal structure that are consistent with its projection onto a subset of variables. In other words, given a causal graphs over different (overlapping) sets of variables, we would like to construct a causal graph over the union of the variables consistent with the given graphs (i.e., conditional independence). Existing sound and complete algorithm ION (Tillman et al. 2008) is computationally efficiently formulated by the authors by employing answer set programming (ASP), solved with an ASP system called clingo. The authors provided soundness and completeness of their approach and simulation results with varying graphs.

**Strengths:**

- The problem itself is well-motivated and Sec 1 introduction to Sec 3 problem setting and method are easy to follow.
- This work revisits an old algorithm and reformulate in a simple clingo problem specification. (I appreciate a simple solution over a unnecessarily complex solution.)
- The combination of ASP and clingo scales better than the original ION algorithm.

**Weaknesses:**

- No new notable theoretical contribution and the main contribution seems rewriting the conditions/constraints in ASP/clingo.

**Questions:**

- What if we run Tillman et al. algorithm in a modern, typical server, how large graphs can be tested instead of 4- and 6-node DAGs? (e.g., evaluate the algorithm in a usual server specification, e.g., 24-CPU cores with 128 GB RAM, etc…)

- What if we re-implement ION taking into consideration of modern hardware (e.g., parallelism, cache, …), do you still think ION-C better than the optimized, specialized implementation of ION? In other words, are there inherent problem with ION or it is just the implementation not being optimized?

**Details Of Ethics Concerns:**

.

---

> ### Author Response · Authors · 2024-11-20
>
> We appreciate the feedback on establishing the contributions of the ASP approach to the ION problem. We have access only to an old implementation of ION, and it is not optimized for modern computational settings. We can appeal to previous work (Hyttinen et al., 2014; Sonntag et al., 2015) on the efficiency of ASP approaches for causal learning problems, in part due to avoiding issues with optimizing implementations, and have gone into more detail about runtime scaling in our simulations with more difficult graphs.

---

> > ### Comment · Reviewer_7BsB · 2024-11-26
> >
> > Thanks for the answers to the questions. Yet, the authors explanations to the reviewers about the novelty or the significance of contribution seem insufficient to change my rating.

---

### Official Review · Reviewer_JqZC · 2024-11-04

**Soundness:** 2
**Presentation:** 2
**Contribution:** 3
**Rating:** 3
**Confidence:** 3

**Summary:**

This paper considers the problem where as input we get a set of overlapping graphs and as output we need to provide all possible DAGs that are consistent with the input graphs according to some rules.

**Strengths:**

The paper uses good English, and shows experimental results suggesting the described goal is achieved of addressing the problem as an ASP problem.   At a high level the proposed approach seems plausible.

**Weaknesses:**

#### General

In general, the paper is probably only readable to specialists.  The paper doesn't provide introductory definitions which would allow a non-expert reader to understand concepts and notations or would allow a more expert reader to disambiguate between definitions of which multiple different ones have been considered in the literature.    See "details below" for a few examples.

The paper doesn't demonstrate clearly what is the added value of representing and solving the problem as an ASP.

The discussion in the paper at several points stays at a high level,
Among others, the experimental section seems to focus on the obtained output but says very little about the runtime cost, even if the problems on which the system is applied look rather small.  Understanding better the scalability may be beneficial.

Overall, while the paper seems potentially interesting, it insufficiently demonstrates the significance of the result and the interest the average ICLR participant may have in it.

#### Details

* L 120: please either provide a detailed definition of ancestral graph or provide a reference where the non-expert reader can find it.  In fact, over time slightly different formalisms and semantics have been proposed, including a clear definition would avoid any ambiguity.
* Footnote 1: I assume "every pair of graphs in the sequence" means "every pair of consecutive graphs in the sequence"
* L 123: Given that the inputs were PAG, indeed they don't need to be DAG.
* I guess "variables" are represented by the vertices in the graphs.
*  "In this problem, there are known latent variables for every input graph (namely, variables that are only in a different graph).".  This is unclear: if a variable (vertex) $v$ is in an input graph $G_1$, how can it be "only in a different graph" ? "only" suggests that $v$ is not in $G_1$.  I guess you mean "only in a different input graph", as it is easy to construct a different graph containing $v$ or not containing $v$.
* L 129: please provide a definition of or reference to d-separation & d-connection for the non-expert reader.
* L129: I guess you make implicitly some assumption about the consistency of the input graphs.
* L 131: "Exactly two graphs" may not be fully precise.  Consider the DAG $\{(X,Y),(Y,Z),(X,W),(X,Z)\}$.  This is a DAG and it is consistent with the two original graphs for certain notions of "consistent".  No notion of "consistent" has been defined here, so it is hard to know whether this DAG would be a solution.
* Listing 1 uses clingo, which is useful for those using this system.  To allow a more general population of readers to understand the paper it could help to use a more widely known representation, e.g., logic, even if this would make the listing slightly different from the actual implementation (it is still possible to offer the implementation in supplementary material for reasons of reproducibility).

**Questions:**

--

**Details Of Ethics Concerns:**

--

---

> ### Author Response · Authors · 2024-11-20
>
> > In general, the paper is probably only readable to specialists.
>
> Thank you for the feedback, and we have added additional definitions, and a more precise framing of the problem setting, into the final version of the paper, including the clarifications you provided in the detailed review.
>
> > The paper doesn't demonstrate clearly what is the added value of representing and solving the problem as an ASP.
>
> We have added text to explain several different ways in which the ASP approach can be beneficial. ASP solvers benefit from optimization made in conflict-driven SAT solving, while allowing the problem to be specified incredibly simply, as we show in Listing 1. Existing work in causal learning has indicated that these lead to significant runtime improvements for ASP approaches (Hyttinen et al., 2014; Sonntag et al., 2015). In addition, ASP approaches allow existing knowledge/beliefs about the ground-truth to be easily specified with simple constraints, whether soft or hard.
>
> > it could help to use a more widely known representation, e.g., logic
>
> We include the clingo implementation due to its relative simplicity, but we have tried to improve the clarity of Listing 1 by being more detailed about how each clingo constraint corresponds to the higher-level description provided in Lines 150-186. We also now describe the clingo syntax further, particularly the different meanings of the `:-` operator in each line.

---

### Meta-Review · Area_Chair_ADRy · 2024-12-19

**Metareview:**

The paper considers the setting where the domain is represented with several datasets, each involving a subset of the domain variables. Continuing the work of Tillman et al. 2008, (ION), the authors propose the ION-C approach, where the problem (of finding the causal graphs consistent with the partial graphs found from each dataset) is formulated as an answer set programming problem, leading to enumerate all causal graphs consistent with the constraints (e.g. independence) issued from the local graphs.

**Additional Comments On Reviewer Discussion:**

The reviewers' concerns regard the added value of the contribution (in theoretical or computational terms) and the readability of the paper for non-experts.
In particular, as put by reviewer 7BsB, it is unclear whether the gain of ION-C wrt ION can be attributed to the lack of optimization of ION.
The authors' rebuttals did not adequately address these concerns.

---

### Decision · Program_Chairs · 2025-01-22

Reject